# *Eucalyptus* Are Unlikely to Escape Plantations and Invade Surrounding Forests Managed with Prescribed Fire in Southeastern US

Fábio Henrique Toledo [1,*], Tyler McIntosh [1], Candice Knothe [2] and Douglas P. Aubrey [1,3]

[1] Savannah River Ecology Laboratory, University of Georgia, Aiken, SC 29802, USA;
Tyler.McIntosh@uga.edu (T.M.); daubrey@uga.edu (D.P.A.)
[2] Polk County Parks and Natural Resources, Lakeland, FL 33803, USA; CandiceKnothe@polk-county.net
[3] Warnell School of Forestry and Natural Resources, University of Georgia, Athens, GA 30602, USA
[*] Correspondence: fhtoledo@uga.edu; Tel.: +1-803-725-0070

**Abstract:** Woody biomass production can increase through establishment of non-native tree species exhibiting greater growth potential than traditional native species. Interest in growing *Eucalyptus* in the southeastern US has raised concern over its potential spread and invasion, which could impact ecosystem properties and functions. Within the matrix of land use where *Eucalyptus* establishment is being considered in the southeastern US, surrounding pine forests managed with fire represent a likely pathway for invasion. We used greenhouse and field experiments to evaluate the potential invasion risk of *Eucalyptus benthamii*. We were specifically interested in determining if seeds could successfully germinate in fire-maintained pine forests and if fire-return intervals influenced germination through impacts on litter accumulation and light availability. The greenhouse experiment investigated the influence of light availability on germination success, whereas the field study investigated the influence of time since fire, and thus litter accumulation and light availability, on germination success. Percent germination was similar under non-shaded controls and moderate shade, but complete shade resulted in low germination rates. Germination was lower in the field compared to the greenhouse and was influenced by litter and light availability, which varied according to fire-return intervals. Litter increased, and light availability decreased, with time since burn. Germination was negatively related to litter depth and positively related to light availability, thereby decreasing with time since fire. Germination increased with litter removal but remained positively related to light availability after litter removal. Higher germination with litter removal suggests germination is influenced by litter, but higher germination with increased light availability, regardless of raking, suggests germination is also influenced by light availability. Despite these relationships, no seedlings persisted through the growing season. The low germination rates under a variety of field conditions coupled with the lack of persistence suggests establishment may be unlikely, regardless of the surrounding land matrix.

**Keywords:** alien plant; invasive species; planted forest; light availability; germination; litter depth; *Eucalyptus benthamii*

## 1. Introduction

*Eucalyptus* rank among the most economically important trees globally [1]. They exhibit traits, such as fast growth [2,3] and high water use efficiency [4–6], that contribute to their candidacy as an excellent biofuel crop [7]. Most eucalypt species are native to Australia and Indonesia and are adapted to tropical and subtropical environments [8,9]. However, some species have developed frost tolerance through advancements in biotechnology [10] or adaptation [3,9,11]. One species in particular— *Eucalyptus benthamii*—has been shown to exhibit competitive frost tolerance when compared to members of the

same genus [12]. This cold hardiness has allowed *E. Benthamii* production in the southeastern United States, ranging from Texas to South Carolina [13]. However, there is growing concern that the same traits that make *Eucalyptus* a highly productive crop may contribute to its potential to be an invasive species [14].

Non-native species have potential to influence the composition, structure, and function of ecosystems where they are introduced [15,16], and numerous reports suggest *Eucalyptus* species are capable of invading ecosystems across the world [16–18]. For example, after *E. globulus* was introduced and became established (i.e., naturalized) in five different countries with similar Mediterranean climate ranging from shrublands to forests, the understory exhibited decreased height and species richness [19]. Becerra et al. [19] offer several potential mechanisms that led to these results: the litter of *E. globulus* contains an allelopathic compound that inhibits the emergence and growth of other plant species; potentially invasive species often reproduce and grow faster compared to their native competitors; eucalypt species use more water than other understory species and dry them out; light is reduced under the eucalypt canopy relative to control understory sites; and climatic and other abiotic factors affect species interaction within the ecosystem (e.g., precipitation seasonality). Another key threat that invasive species can impose on native ecosystems is the interruption of fire regimes [15]. The leaf litter of *Eucalyptus* is highly flammable [20], and if a stand is left unmanaged, the risk of wildfire to the stand and the surrounding lands is significantly increased. Another consideration is the impact that *Eucalyptus* have on stream flows from catchments of surrounding ecosystems [4]. A few studies on the long-term effects of afforestation on experimental catchments demonstrated that *E. grandis* reduced stream flows relative to native grass cover [21] and *Pinus patula* [22]. The negative impacts on ecosystem composition, structure, and function associated with *Eucalyptus* invasions in other parts of the world suggests their potential escape from plantation in the southeast must be considered.

The potential for *Eucalyptus* to escape forest plantations and invade surrounding ecosystems in the southeastern US remains unclear. According to an assessment using the Australian weed risk assessment tool, several species of *Eucalyptus*, including *E. benthamii*, have the potential to become invasive in the southeastern US [23,24]. However, other reports suggest that, while caution and risk management strategies should be utilized when planting eucalypt species in non-native areas, the potential for these species to become established in, and invade surrounding areas is relatively small [2]. Rejmánek and Simberloff [20] suggest three reasons why *Eucalyptus* spp. exhibit a relatively low invasion risk: limited seed dispersal, high mortality rate of seedlings, and potentially low colonization by ectomycorrhizal fungi that may be essential to seedling establishment. An experiment conducted by Calviño-Cancela and Rubido-Bará [25] demonstrated that *E. globulus* seeds exhibit relatively low dispersal distances, with most seeds (84.3%) falling within five meters of the parent trees. Furthermore, *Eucalyptus* seeds do not have an obvious endosperm, so they rely heavily on post-emergence cotyledon photosynthesis for sustenance [20]. This suggests that *Eucalyptus* seedlings have an immediate need for light and water, which implies that seeds are more likely to germinate on wet, bare soils free of litter [20].

Successful invasion by *Eucalyptus* ultimately depends on seed dispersal, seed germination, and seedling establishment [26]. Assuming *Eucalyptus* seed dispersal is adequate, successful germination and establishment may depend on land management practices of areas surrounding plantations that influence site characteristics, such as soil moisture [18,27], light availability [25], and litter accumulation [25]. It has been suggested that there is low likelihood for *Eucalyptus* spp. to escape plantation sites and invade surrounding areas, especially north of latitude 27° N, but the degree of likelihood may depend on the land use of those surrounding areas [28]. In the southeastern US where the matrix of land use is broadly comprised of croplands, grasslands, urban land, and forested land [29], the most likely potential pathways for invasion involve spread of *Eucalyptus* seed from plantations to surrounding agricultural fields, old fields (i.e., unmanaged), pine plantation forests, and pine forests managed with prescribed fire (Figure 1). Agricultural land uses include heavy land management tactics, such as herbicide application and species control [30], that are not conducive to the

establishment of non-native tree species. Forest-use lands and less extensively managed fields exhibit increased leaf litter and decreased light penetration [31,32], which are also not conducive to non-native tree establishment. *Eucalyptus* species are more likely to become invasive in lands that are managed with prescribed burns [33–35]. Following prescribed fire, conditions such as reduced vegetation [33], reduced leaf litter depth [33], increased light availability to the forest floor [33], and altered litter chemistry [36] might provide suitable physical and biological conditions for seed germination and seedling development.

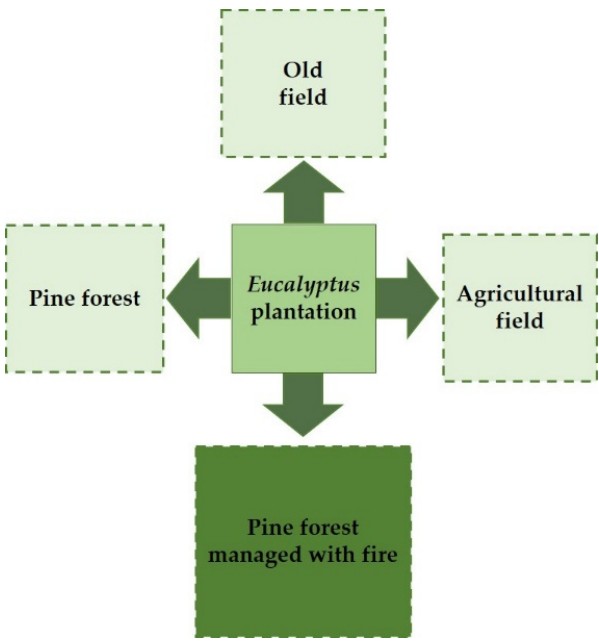

**Figure 1.** Potential pathways for *Eucalyptus* invasion in the southeastern US.

In this study, we examined the potential for *E. benthamii* to become established in pine forests managed with prescribed fire using a combination of greenhouse and field studies focused on *E. benthamii* germination as a function of light availability and litter accumulation. Our objective with the greenhouse study was to compare the effect of shading on the germination rate of the *Eucalyptus* species. Our objective with the field study was to compare seed germination rates under different levels of canopy light penetration and different leaf litter depths resulting from different times (0, 2, and 4 years) since the prescribed fire. We predicted that canopy light penetration would decrease, and leaf litter depth would increase, with increasing time since fire. Based on this presumed response, we predicted that seed germination rates would decrease with increasing time since fire.

## 2. Materials and Methods

### 2.1. Greenhouse Study

We evaluated *Eucalyptus benthamii* seed germination rates in response to different levels of shade (0, 60, and 100% shade) in a greenhouse by using a plastic covering, shade cloth, or a combination of these two placed over the top and side of cone-tainers™. The 0% shade treatment was imposed by using 0.068 mm thick clear plastic material and placing it over the top and sides of cone-tainers™. The 60% shade treatment was imposed by combining 0.068 mm thick clear plastic material and 60% polyethylene netted shade cloth (Sunblocker, Dyersville, IA, USA). The 100% shade treatment was imposed by using 0.076 mm thick black plastic material (contractor garbage bag). Thus, all plastic covering was of similar thickness (0.068–0.076 mm) to maintain consistent humidity across shade treatments. We added Metromix 360 commercial potting mixture (Sun Gro Horticulture, Agawam, MA, USA) to SC-10 Super Cell Ray Leach cone-tainers™ (diameter, 3.8 cm; depth, 21 cm; Stuewe &

Sons, Corvallis, OR, USA) placed in trays (17.2 cm height; Stuewe & Sons). Photosynthetically active radiation (PAR) was measured using a quantum sensor to validate percent shading before the start of the study (LI-250A, LI-COR Biosciences, NE, USA). Shade treatments were imposed immediately after the seeds were sown and maintained for 25 days. *E. benthamii*, seeds are extremely small (2–4 mm in length) and contained within woody capsules [37]. Seeds used in this study were obtained from ArborGen, Inc. (Ridgeville, SC, USA).

The experiment was conducted in a randomized block design with 3 treatments (shade levels) and 12 replications (blocks), totaling 36 plots. Each plot comprised one tray with 25 cone-tainers™. We sowed one *E. benthamii* seed on the surface of the soil of each cone-tainers™ of each tray. We watered cone-tainers™ to soil saturation daily and maintained temperature (21–26 °C) and air humidity (70%) in the greenhouse room throughout the experiment. We assessed germination visually (by any extension of the radicle, hypocotyl, or root) two days after seeds were sown, and then every four days for one month. We removed and replaced shade covering to assess germination. We applied additional water as needed during germination assessments.

*2.2. Field Study*

We evaluated *Eucalyptus benthamii* seed germination rates in response to different time since burn and litter removal treatments at the United States Department of Energy's Savannah River Site (SRS), located on the upper Atlantic Coastal Plain [38]. This approximately 80,000 ha site is a National Environmental Research Park located in parts of Aiken, Barnwell, and Allendale Counties, South Carolina USA (33° 18' N, 81°37° W). The region has a humid subtropical climate characterized by relatively short, mild winters and long, warm and humid summers [39]. Average temperatures in the winter are 9 °C and 26 °C in the summer [38]. Precipitation averages 1225 mm a year [39]. Seven soil associations are found on the site, the majority being sandy silt, occupying uplands and ridges, and loamy clays occupying streams and flood plains [38]. Approximately 90% of the area that comprises SRS consists of natural and managed forests [39]. Pine forests dominate the landscape which consists of a patchy mosaic of managed, even-aged longleaf pine (*Pinus palustris*) and loblolly pine (*Pinus taeda*) stands of varying age-classes [38,39]. Approximately 6000 to 8000 ha of pine forests are managed with prescribed fire annually, with targeted burn frequencies averaging between 2–4 years [40]. See Aubrey et al. [41] for a detailed description of the site and its recent forest history.

We used the Savannah River Forest Service's maps and prescribed fire records to locate loblolly pine forest stands that had received prescribed fire (i.e., low-intensity surface fire [42]) during the winter or spring prior to our study (i.e., 0 years since burn), two years prior to the study (i.e., 2 years since burn), or four years prior to the study (i.e., 4 years since burn). This range of burn history characterizes variation in fire-free intervals and resulting differences in litter accumulation and canopy characteristics that occur in pine forests across the southeastern landscape.

We established eight 60 × 60 m plots within each time since burn treatment (0, 2, or 4 years since burn) for a total of 24 experimental plots. We defined whole plots by the time since burn treatment. Within each plot, we randomly established three subplots that were treated as split-plots, where we randomly assigned one of two litter removal treatments (control or litter removal; hereafter referred to as rake treatment) to each half. Litter removal could affect the micro-climate in different ways since litter helps maintain soil moisture, but also limits a portion of solar radiation reaching the ground. We used a metal rake to remove all ground litter and expose bare soil in the subplot receiving the rake treatment, whereas we left litter intact for the subplot receiving the control treatment. We established parallel transects separated by 1.5 m within each half of the subplot. We placed pin flags every 0.5 m along each transect to designate seed sowing locations. Before applying the litter removal treatment, we measured litter depth every 1.5 m along each transect for a total of six litter depth measurements at each subplot and calculated the average litter depth for the whole plot across those subplot measurements. We used the same spatial sampling protocol to measure the quantity of light penetrating the forest canopy (i.e., canopy light penetration) and the quantity of light penetrating the

leaf litter (i.e., litter light penetration). We measured photosynthetically active radiation (PAR) using a ceptometer (AccuPar LP-80, METER Group, Inc., WA, USA) and present data as a percentage of open canopy PAR, which was measured in nearby clearings. To measure the litter light penetration, we carefully insert the ceptometer between the litter layer and the soil surface, while the canopy light penetration was measured positioning the ceptometer at the breast high (1.3 m above the soil).

We sowed individual *E. benthamii* seeds using forceps and placed them on the soil surface directly 1 cm in front of each pin flag along each transect in the subplots. We supplied 20 mL of water directly to the soil under the seed at the time of planting and an additional 20 mL of water two days after sowing. No water addition was needed after that due to the weather conditions (Figure S1). We planted a total of 10 seeds in each subplot. We determined germination visually following the same approach used in the greenhouse experiment. We conducted germination surveys from 11 July to 10 December. We conducted surveys every four days for the first two weeks after seeds were sown. As germination declined over time, our surveys were conducted weekly and then monthly for the remainder of the study. We conducted a final survey 340 days following sowing.

### 2.3. Data Analysis

We used analysis of variance (ANOVA) to compare treatment effects. Percent germination data were arc sine transformed prior to analyses to achieve normality [43], but untransformed means and standard errors are presented in the figures. A randomized block design was used to analyze percent germination in the greenhouse study. The shade treatment (0, 2, or 4 years since prescribed fire) was treated as a fixed factor and the experimental unit-by-block, individual plot, was treated as a random factor. A split-plot design was used to analyze percent germination in the field study. Prescribed fire regime (i.e., 0, 2, or 4 years since prescribed fire) was treated as the fixed whole-plot factor and litter removal (i.e., raked or unraked) was treated as the fixed sub-plot factor. A similar model was used to analyze litter depth, canopy light penetration, and litter light penetration; however, the litter depth and litter light penetration models did not include the raked sub-plot, just the years since prescribed fire whole-plot factor. The ANOVAs were performed using the mixed model procedure (PROC MIXED) of SAS 9.4 (SAS, INC., Cary, NC, USA) with $\alpha = 0.05$ and Tukey's honestly significant difference test (HSD) was used for mean comparisons.

We used linear regression to explore relationships between seed germination percentages and forest characteristics, such as litter depth, canopy light penetration, and litter light penetration. When exploring relationships between percent germination and litter depth or litter light penetration, only data from unraked sub-plots were used. Similarly, when exploring relationships between percent germination and canopy light penetration, only data from raked sub-plots were used. Linear regressions were performed using the linear regression procedure (PROC REG) of SAS 9.4

## 3. Results

### 3.1. Greenhouse Germination

The shade treatment influenced germination rates ($p < 0.05$) (Figure 2). The 100% shade treatment resulted in statistically lower germination rates (15.7 ± 1.0%) relative to the 0% (74.7 ± 1.0%) and 60% shade (70.7 ± 1.7%), which were statistically similar. The shade treatment did not influence time to germination ($p > 0.05$) (data not shown). The mean time to germination across all shade treatments was 8.12 ± 0.14 days after seeds were sown. Germination was first observed 5 days after sowing and no additional germination occurred 21 days after seeds were sown.

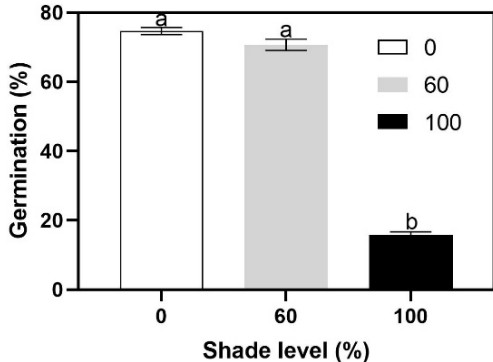

**Figure 2.** *Eucalyptus benthamii* seed germination under three levels of shade in the greenhouse. Error bars indicate the standard error. Means sharing a letter are not significantly different (Tukey's HSD, $\alpha$ = 0.05).

### 3.2. Field Study

Leaf litter depth, canopy light penetration, and litter light penetration were influenced by year since burn treatments. Leaf litter depth increased with increasing time since burn ($p$ = 0.0112). The 4-years since burn treatment exhibited the highest litter depth and the 0-years since burn treatment exhibited the lowest litter depth (Figure 3a). The 2-years since burn treatment exhibited a litter depth that was intermediate and did not differ from the extremes (Figure 3a). Canopy light penetration and litter light penetration both decreased with increasing time since burn (both $p$ < 0.0001). Canopy light penetration and litter light penetration were highest in the 0-, intermediate in the 2-, and lowest in the 4-years since burn treatments (Figure 3b,c). The amount of light penetrating the leaf litter layer was approximately one-half or less than the amount of light penetrating the canopy.

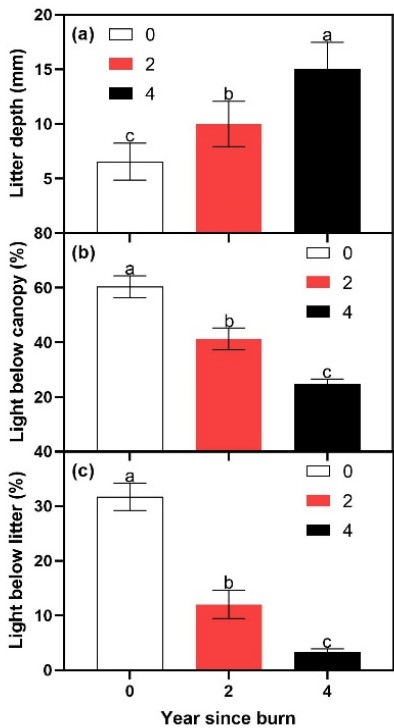

**Figure 3.** Litter depth (**a**) and percentage of light below canopy (**b**) and below litter (**c**) after zero, two, and four years after a prescribed fire. Error bars indicate the standard error. Means sharing a letter are not significantly different (Tukey's HSD, $\alpha$ = 0.05).

Overall, germination rates were much lower in the field compared to the greenhouse. The year since burn and raking treatments did not influence time to germination ($p$ > 0.05). The mean time to

germinate across all year since burn x litter removal treatments was 7.45 ± 0.17 days after seeds were sown. Germination was first observed 5 days after sowing and no additional germination occurred more than 15 days after seeds were sown.

Germination rates were influenced by the year since burn and raking treatments (Figure 4). The year since burn effect was independent of raking ($p$ = 0.0008), but the raking effect was dependent on the year since burn treatment (i.e., year since burn × raking interaction; $p$ = 0.0371). Germination rates decreased with increasing time since burn. Those were highest in the 0-, intermediate in the 2-, and lowest in the 4-year since burn treatments. Germination rates increased with raking, but the magnitude of difference between raked and unraked controls was not consistent across the years since burn treatments.

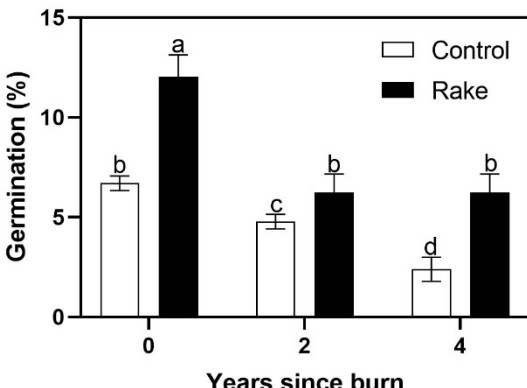

**Figure 4.** Percentage of *Eucalyptus benthamii* seed germination after zero, two, and four years after prescribed fire with and without rake in the field. Error bars indicate the standard error. Means sharing a letter are not significantly different (Tukey's HSD, $\alpha$ = 0.05).

Differences in germination rates among the time since burn and raking treatments could partially be explained by litter depth, litter light penetration, and canopy light penetration (Figure 5). Germination rates in unraked treatments decreased with increasing leaf litter depth (Figure 5a) and increased with increasing litter light penetration (Figure 5b), whereas germination rates in raked treatments increased with increasing canopy light penetration (Figure 5c). Seedling mortality was first observed 21 days after sowing. No seedlings remained 97 days after sowing and a final survey conducted 340 days after sowing found no reappearance of seedlings or additional germination.

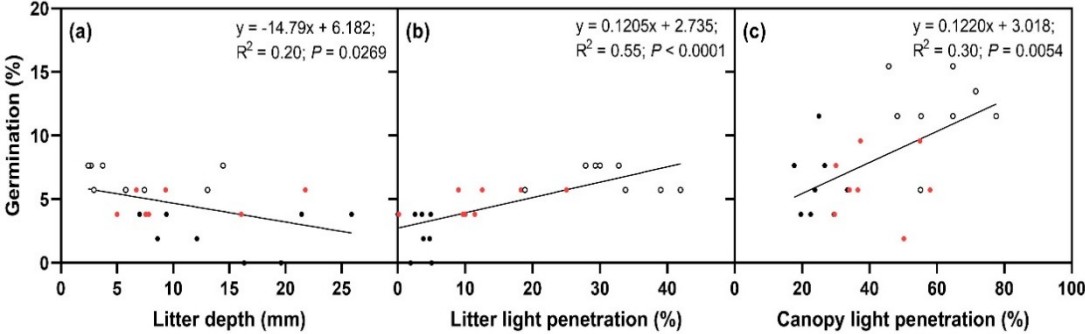

**Figure 5.** Correlation of *Eucalyptus benthamii* seed germination with litter depth (**a**) litter light penetration (**b**) and canopy light penetration (**c**) after zero, two, and four years after prescribed fire. White, red, and black dots are 0, 2 and 4 years since burn, respectively. When exploring relationships between percent germination and litter depth (**a**) or litter light penetration (**b**), only data from unraked sub-plots were used. Similarly, when exploring relationships between percent germination and canopy light penetration (**c**), only data from raked sub-plots were used.

## 4. Discussion

Results from our greenhouse and field experiments provide important information regarding the potential for *Eucalyptus benthamii* to become established in pine forests managed with prescribed fire. Our greenhouse experiment demonstrated that, under ideal moisture and temperature conditions, percent germination of *E. benthamii* seeds was similarly high under non-shaded controls and moderate shade; however, complete shade resulted in much lower germination rates. As we predicted, canopy light penetration decreased, and leaf litter depth increased with increasing time since fire. Also as predicted, seed germination rates decreased with increasing time since fire, exhibiting a negative relationship with leaf litter depth and a positive relationship with light availability. However, overall germination rates were much lower in our field experiment compared to our greenhouse experiment. Germination increased with litter removal but the positive relationship between germination and light availability was also present after litter removal. Higher germination with raking suggests germination is influenced by litter depth (Figure 4), but higher germination with increased light availability, regardless of raking, suggests germination is also influenced by light availability (Figure 5c). Despite these relationships, no seedlings persisted through the growing season. The low germination rates under a complex field conditions, coupled with the absence of seedling development, suggests establishment may be unlikely, regardless of the surrounding land matrix.

Prescribed fire had a fairly predictable effect on forest structure. Litter depth increased every two years after burning (Figure 3a), which is similar to what is typically observed following prescribed fire in a variety of ecosystems [31,44,45]. For example, 50% of pre-burn leaf litter had accumulated within three years of prescribed fire and 100% had accumulated within six years of prescribed fire in southern pine ecosystems [31]. Canopy light penetration decreased every two years after burning (Figure 3b), which is similar to what is typically observed following prescribed fire in a variety of ecosystems [46]. Not surprisingly, the litter light penetration followed the inverse pattern shown by the litter depth and the same pattern shown by the percentage of canopy light penetration (Figure 3c). Germination rates of *E. benthamii* were, in turn, influenced by the changes in litter depth, canopy light penetration, and litter light penetration that differed among the time since burn treatments.

Our study showed that the time to germination of *E. benthamii* in the greenhouse (~8 days) was not influenced by the percentage of shade. Previous studies showed that time to germination is more influenced by micro-climate conditions such as soil moisture and temperature [47]. In our greenhouse study, soil moisture and air temperature were both controlled to maintain favorable conditions to promote germination. Aligned to that, as *E. benthamii* seeds are small and lack endosperm, a fast germination rate is part of the species' survival strategy [18,48,49]. Indeed, we observed similar times to germination in our field study (~7 days) under a very different set of environmental conditions than the greenhouse study, suggesting that general traits related to *E. benthamii*'s germination strategy exert stronger controls than do environmental factors. Similar times to germination have been observed for *E. globulus* [33].

We observed that 0 and 60% shade promoted similarly high germination rates under ideal conditions in the greenhouse, but 100% shade significantly reduced germination rates (Figure 2); however, we could not identify a critical threshold for light availability since we did not test a broader continuum of light availabilities. Although the absence of light directly affected germination rates in our study, other studies have concluded that low light availability may not impede germination, but instead impedes seedling development [33,50]. Calviño-Cancela et al. [33], testing *E. globulus* germination, found that seeds under suspended litter and plastic trellis (85% to 100% of light interception) developed fewer leaves or no leaves at all until they died. In our greenhouse study, we only considered the shading effect on germination, not on seedling development. *Eucalyptus* spp. seed germination in the field experiment were drastically lower than in controlled conditions. In fact, the lowest germination rates observed in our greenhouse study under 100% shade (15.7%) were similar to the highest germination rates observed in the field (15.1%). Seeds in our 2-years since burn treatment that were raked to remove litter experienced 41.3% canopy light penetration, which is similar to the 40% light availability in the

greenhouse experiment (i.e., 60% shade); however, germination rates in the field were 6.2% compared to the 70.7% in the greenhouse (Figures 2 and 5c). Similar germination responses to light availability have been observed in other studies and for other eucalypts species [27,33]. Calviño-Cancela et al. [33] found germination rates varying from ~85% in the greenhouse to ~22% in *Quercus robur* forest burn sites and concluded that litter negative effect is also related to its physical impediment to seedling emergency. In natural conditions, the factors that can potentially limit seed germination and emergence are water stress [27,47], including droughts [47], cold temperatures [47], litter depth [33], susceptibility to pathogens and herbivory [27,33], and sensitivity to plant competition [36,51]. Non-germinating seeds were identified, so herbivory did not influence our results. Although we have no way of assessing pathogens, we do not believe this to be an important factor limiting germination in our study. Likewise, temperature should not have been an influential factor. Our methodological approach of supplying water to seeds the day of sowing and again two days later should have mitigated any soil resource limitations for germination. Moreover, we did not continue supplementing water beyond the two days after sowing because precipitation events improved soil moisture over the critical time period for germination (i.e., >20 days after sowing; Figure S1). Therefore, our results suggest that differences in germination rates were largely due to differences in light availability that resulted from differences in time since prescribed fire.

By measuring litter depth, canopy light penetration, and litter light penetration, we were able to explore how changes in forest structure that result from prescribed fire and the time since prescribed fire can influence *E. benthamii* germination. For example, our study shows that litter removal, regardless of time since burn, has a beneficial effect on germination (Figure 3). The absolute effect of litter removal was strongest in the 0-years since burn treatment (6.7% vs. 12.0% for control and rake, respectively), but the proportional increase was highest in the 4-years since burn treatment (2.4% vs. 6.2% for control and rake, respectively). Our litter removal treatment helps identify how other forest stand characteristics that can be influenced by prescribed fire also influence germination rates of *E. benthamii* seeds. For example, the large difference in germination rates between the 0-year since burn and the 2-year since burn treatments with litter removal, but similar germination rates in the 2-year and 4-year since burn treatments with litter removal suggests that canopy light penetration limits germination rates a few years after burning. Likewise, the decrease in germination rates between the 2-years since burn and the 4-years since burn treatments without litter removal suggest the combination of canopy light availability and litter depth influence germination rates. Our results corroborate with the ones found by Fernandes et al. [52], who in the study of *E. globulus* seed germination after soil disturbances, found higher germination in raked (~34%) than in unraked areas (~19%). Although there may be times when leaf litter facilitates germination because of its ability to hold moisture [32,53], the relationships we observed between germination rates and light availability (both canopy and litter light penetration) suggest that leaf litter is more of a hindrance to seed germination.

## 5. Conclusions

Based on our findings, we conclude that *Eucalyptus benthamii* exhibits low potential of becoming established outside of plantations in pine forests managed with prescribed fire in the southeastern US. High germination rates occur only under ideal conditions, and our field experiment demonstrated that litter depth has a negative impact, whereas light availability (through litter or canopy) has a positive impact on germination rates in the field. Regardless of responses to litter depth and light availability, germination rates in the field were extremely low. Critically, not a single seedling persisted through the initial growing season. The low germination rates under a variety of field conditions coupled with the lack of persistence suggests establishment may be unlikely, regardless of the surrounding land matrix. Similar to our results, Lorentz and Minogue [2] studying *E. grandis*, *E. camaldulensis*, and *E. amplifolia* survival found that 86% of the seedlings survived less than one month, and no seedling survived more than 13 weeks in two different locations in Florida.

Even with low invasion risk [24] *E. benthamii* can become invasive due to intrinsic population growth, selection for tolerance to new environments, and change in biotic and abiotic characteristics (e.g., climate change) [54]. As the lag time of the introduction of a specie and its identification as invasive is 170 years, on average, those factors can act for a long time [24,54] until it finds the perfect conditions for naturalization. Thus, we emphasized that plantations of *E. benthamii*, as well as other species and clones of the genus, should be conducted cautiously following best management practices that reduce invasion risk, such as (i) avoiding cultivation near waterways [9,24]; (ii) managing plantations to reduce propagule pressure (e.g., short rotation to avoid seed maturation) [24,28]; (iii) maintaining clear firebreaks [24]; (iv) planting sterile genotypes or clones for low levels of seed production [9,24]; (v) planting and maintaining plantations with low within-population genetic variability [55].

**Supplementary Materials:** The following are available online at http://www.mdpi.com/1999-4907/11/6/694/s1, Figure S1: Precipitation and soil moisture during the field experiment.

**Author Contributions:** Conceptualization, D.P.A. and C.K.; methodology D.P.A. and C.K.; statistical analysis D.P.A. and C.K.; writing—original draft preparation, F.H.T., T.M., and D.P.A.; writing—review and editing, F.H.T., T.M., and D.P.A.; funding acquisition, D.P.A. All authors have read and agreed to the published version of the manuscript.

**Funding:** This work was supported by the USDA National Institute of Food and Agriculture, Agriculture and Food Research Initiative (grant numbers 2013-67009-21405, 2013-67009-25148, and was based upon work supported by the Department of Energy to the University of Georgia Research Foundation (grant number DE-EM0004391) and to the U.S. Forest Service Savannah River (grant number DE-EM0003622).

**Acknowledgments:** We thank Jennie Haskell and Linda Lee for assistance with locating field sites.

**Conflicts of Interest:** The authors declare no conflict of interest.

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
