# Peer review of "Eucalyptus Are Unlikely to Escape Plantations and Invade Surrounding Forests Managed with Prescribed Fire in Southeastern US"

_forests, doi:10.3390/f11060694_

Round 1
Reviewer 1 Report
Paper deals with introduced tree species Eucalyptus benthamii and his ecological and growing properties, supported by greenhouse- and field experiments. So its opportunities and environmental risks. My my opinion is in the recent time research targeted on escaping of non-native and invasive species and next invade surrounding ecosystems globally very opportune. I consider this topic as actual and relevant. Chapter Abstract and Introduction are by my opinion good prepared. To presented work, I have following comments:
Material and Methods
Row 118 – instead “27 mil” I understand “27 mm thick “clear plastic material… please, correct too in next text
I consider selected levels of shade (0%, 60%, 100%) as questionable. By my opinion, better level in experiment observation would be next to 0% and 100%, treatment with e.g. 25-30% shade, 45-50% shade and/or 70-75% shade. 100% shade I consider as unnecessary, in the natural habitats is this event very rarely. But I respect, that after establishment of experiments some years ago is this recommendation not relevant. I agree with proposed design, 36 plots (each with 25 containers – please see correct writing “conetainers” in row 132) in greenhouse study and 24 plots with dimensions 60x60 m in the field with 3 burn treatments (0, 2 or 4-years since burn) and 2 litter removal treatment (control and litter removal). Because of greater variability, I recommend use more plots for Field experiment, but I respect economical aspect. Establishment of experiments and too data analysis I consider as comprehensive. But I not enough understand how was the quantity of light penetrating the leaf litter. Please specify localization of ceptometer, it was putting under litter layer on the soil background? Or?
Results
Figure 2 results from proposed methods, I consider figure as little insufficient. To correct generalization I lack more information’s about shade between 0 and 60%. However, from proposed experiments is it impossible. I see none differences between 0 and 60% shade level; germination is in general very high. Practical output by my opinion for observed habitats is very good germination. By question or research of invasions potential of Eucalyptus must follow data survey of living and growing of Eucalyptus regeneration.
As very relevant, I consider results in Figure 4 and Figure 2, so differences between Greenhouse and Field experiment. Figures presents limited potencial in the Field compared in Greenhouse. In Figure 5 I recommend append legend (I think red points are plots 2 years since burn and black points 4 years after burn, but you must explain in figure or explanatory text).
I consider presented work as little weak, but I think that authors made respectable research and have come to an interesting end. Their research outputs would be interesting for others researchers and I recommend in spite a few my comments after a some changes and modifications for publishing in journal Forests.
Reviewer 2 Report
This paper is the result of a well-designed and executed project. The results should be of interest to forest science and natural resource management communities.
The paper is well written and flows well. I only spotted two or three typos on species names and other words (see attached pdf with sticky notes and highlights).
The Discussion and Conclusions are well written and appropriately 'hedge' on applicability.
One does wonder if other land uses pose greater escape and invasion risk, including road corridors and urban/suburban land. But the authors seem to make appropriate references to this possibility and related risk mitigation options.
